# A Molecular Genetic Analysis of RPE65-Associated Forms of Inherited Retinal Degenerations in the Russian Federation

**DOI:** 10.3390/genes14112056

**Published:** 2023-11-09

**Authors:** Anna Stepanova, Natalya Ogorodova, Vitaly Kadyshev, Olga Shchagina, Sergei Kutsev, Aleksandr Polyakov

**Affiliations:** Research Centre for Medical Genetics, Moscow 115478, Russia

**Keywords:** *RPE65*, inherited retinal degenerations, RPE65-associated retinopathies, c.304G>T/p.(Glu102*), c.370C>T/p.(Arg124*), c.272G>A/p.(Arg91Gln), common mutations, *RPE65* mutation spectrum, uniparental isodisomy

## Abstract

Pathogenic variants in the *RPE65* gene cause the only known form of inherited retinal degenerations (IRDs) that are prone to gene therapy. The current study is aimed at the evaluation of the prevalence of RPE65-associated retinopathy in the Russian Federation, the characterization of known variants in the *RPE65* gene, and the establishment of the specificities of the mutation spectrum in Russian patients. Methods: The analysis was carried out on blood samples obtained from 1053 non-related IRDs patients. The analysis, which consisted of 211 genes, was carried out based on the method of massive parallel sequencing (MPS) for all probands. Variant validation, as well as biallelic status verification, were carried out using direct automated Sanger sequencing. The number of copies of *RPE65* exons 1–14 was analyzed with quantitative MLPA using an MRC-Holland SALSA MLPA probemix. Results: Out of 1053 non-related patients, a molecular genetic diagnosis of IRDs has been confirmed in 474 cases, including 25 (5.3%) patients with RPE65-associated retinopathy. We detected 26 variants in the *RPE65* gene, nine of which have not been previously described in the literature. The most common mutations in the Russian population were c.304G>T/p.(Glu102*), c.370C>T/p.(Arg124*), and c.272G>A/p.(Arg91Gln), which comprised 41.8% of all affected chromosomes. Conclusions: The current study shows that pathogenic variants in the *RPE65* gene contribute significantly to the pathogenesis of IRDs and comprise 5.3% of all patients with a confirmed molecular genetic diagnosis. This study allowed for the formation of a cohort for target therapy of the disorder; such therapy has already been carried out for some patients.

## 1. Introduction

Inherited retinal degenerations (IRDs) are a group of heterogeneous genetic disorders leading to visual impairment caused by progressive retinal degeneration.

The prevalence of IRDs in Europe is approximately 1:3000 [1,2,3]. IRDs comprise multiple rare disorders with varying ages of manifestation, rates of progression, primary retinal damage levels, and inheritance types. The symptoms of retinal dystrophy may vary from congenital blindness, in cases of more severe forms of retinal degeneration, to mild retinal dysfunction manifesting as night blindness. As of date, there are more than 340 known genes causing nonsyndromic and syndromic retinal diseases with autosomal recessive, autosomal dominant, and X-linked types of inheritance (according to https://sph.uth.edu/retnet, 12 September 2023).

Considering the prevalence of IRDs, in the Russian Federation (population of approximately 145 million), there are more than 48,000 people suffering from various forms of RP. Although the majority of the disorders are quite rare, in aggregate, they are a common cause of blindness or significant vision impairment in infants and adults of working age.

RPE65-associated retinopathies became the first group of hereditary ophthalmological disorders with approved gene therapy; therefore, the *RPE65* gene came under the close attention of researchers from around the world.

The *RPE65* gene is located on the 1p31 chromosome, contains 14 exons, and encodes proteins with a mass of 65 kDa, which consists of 533 amino acid residues [4]. RPE65 is a membrane-associated protein, which is expressed primarily in the pigment epithelium and plays a vital role in the regeneration of 11-cis-retinol in the visual cycle [4]. Light activates the visual pigment, causing it to break down hydrolytically with the formation of a free all-trans-retinal, which regenerates in the external segments of photoreceptors using retinol dehydrogenase. The resulting all-trans-retinol is transferred to the pigment epithelium, where it is transformed into all-trans-retinyl esters with retinol acyltransferase [5]. The all-trans-retinyl esters can be stored in retinosomes or be further converted [6]. In the retinal pigment epithelium, RPE65 converts all-trans-retinyl esters into 11-cis-retinol and a free fatty acid via simultaneous hydrolysis and isomerization reactions, thereby being named isomerohydrolase. After that, in the pigment epithelium, the 11-cis-retinol is oxidized using 11-cis-retinol dehydrogenase (RDH5) to 11-cis-retinal [7], then returns to the external segment of the photoreceptor to regenerate main visual pigments, thus completing the cycle [8].

RPE65 alterations lead to the accumulation of all-trans-retinyl esters and a decrease in or absence of visual pigment in the photoreceptors.

A fully active RPE65 protein is a dimer of two symmetrical enzymatically independent subunits. A monomer of RPE65 is a seven-bladed β-propeller and an iron-dependent enzyme, in which four histidine residues (His180, His241, His313, and His527) and three glutamic acid residues (Glu148, Glu417, and Glu469) coordinate an Fe^2+^ cation [9].

Biallelic pathogenic variants in the *RPE65* gene cause Leber congenital amaurosis (LCA) (OMIM 204100) and severe early-onset retinitis pigmentosa (RP20; OMIM 613794) [10]. In various populations around the world, 0.6% to 16% of IRD cases are caused by pathogenic variants in the *RPE65* gene [4,11]. The most common pathogenic *RPE65* variants differ from population to population; quite often, there are unique variants for specific geographic regions. The most common pathogenic variant among Spanish patients was c.292_311del (p.(Ile98Hisfs*26)) with an allelic frequency of 11% [11]; Greek—c.304G>T (p.(E102*)); Dutch—c.1102T>C (p.(Y368H)); Saudi Arabian and Tunisian—c.271C>T (p.(R91W)); and Danish—c.329A>G (p.(D110G)) [12]. Previously, there were no global studies on the *RPE65* mutation spectrum in the Russian Federation.

Establishing the exact genetic cause of the disorder allows us to predict the course of the disease, evaluate the risk for other family members, and make an offspring prognosis. Moreover, identifying the causative gene is a key step in the possibility of using gene therapy.

## 2. Materials and Methods

An analysis was carried out on blood samples obtained from 1053 non-related patients with a referral diagnosis of “retinitis pigmentosa”, “cone-rod dystrophy”, or “LCA” from various regions of the Russian Federation. In total, 524 (49.8%) patients were female and 529 (50.2%) were male.

All patients underwent standard ophthalmic examinations by ophthalmologists at their place of residence, including Snellen best corrected visual acuity (BCVA), slit lamp biomicroscopy, Goldmann visual field tests, optical coherence tomography (OCT), and full-field electroretinography (ffERG). The criteria for “retinitis pigmentosa”, “cone-rod dystrophy”, or “LCA” diagnosis included decreased vision from early childhood, nystagmus, nyctalopia, atrophic changes in the outer layers of the retina, and electroretinography abnormalities. Probands with any lesions of other systems suggesting a syndromic disease or having affected parents were excluded from the study.

The DNA was extracted from peripheral blood leukocytes using a QIAamp DNA Mini Kit, Qiagen, according to the manufacturer’s protocol.

All proband DNA was analyzed using a custom AmpliSeq™ panel «Ophthalmo» on an Ion Torrent S5 next generation sequencer. The panel «Ophthalmo» includes coding sequences of the 211 genes: *ABCA4*, *ADAMTSL4*, *COL8A2*, *CRB1*, *EPHA2*, *FOXE3*, *GJA8*, *GNAT2*, *HMCN1*, *MYOC*, *NMNAT1*, *PLA2G5*, *PRPF3*, *RD3*, *RPE65*, *SEMA4A*, *UBIAD1*, *USH2A*, *ZNF644*, *ARMS2*, *CDHR1*, *HTRA1*, *OAT*, *OPTN*, *PAX2*, *PDE6C*, *PITX3*, *RAB18*, *RBP3*, *RBP4*, *RGR*, *SLC16A12*, *VIM*, *ZEB1*, *BEST1*, *CABP4*, *CAPN5*, *CEP164*, *CRYAB*, *FZD4*, *LRP5*, *MFRP*, *PAX6*, *ROM1*, *TEAD1*, *TMEM126A*, *ZNF408*, *CEP290*, *DCN*, *GDF3*, *KERA*, *KIF21A*, *KRT3*, *MIP*, *PDE6H*, *RDH5*, *GJA3*, *GRK1*, *RB1*, *LTBP2*, *NRL*, *OTX2*, *RDH12*, *SIX6*, *SMOC1*, *SPATA7*, *TTC8*, *VSX2*, *ALDH1A3*, *NR2E3*, *OCA2*, *POLG*, *SLC24A1*, *STRA6*, *TRPM1*, *ABCC6*, *ARL2BP*, *BBS2*, *CHST6*, *CLN3*, *CNGB1*, *HSF4*, *MAF*, *SLC38A8*, *TUBB3*, *AIPL1*, *CA4*, *CRYBA1*, *FSCN2*, *GPR179*, *GUCY2D*, *KRT12*, *PDE6G*, *PITPNM3*, *POLG2*, *PRCD*, *PRPF8*, *RGS9*, *UNC45B*, *RAX*, *TCF4*, *CRX*, *LIM2*, *NTF4*, *OPA3*, *PRPF31*, *RGS9BP*, *SIPA1L3*, *ABCB6*, *C2orf71*, *CERKL*, *CHN1*, *CNGA3*, *CNNM4*, *CRYBA2*, *CRYGC*, *CRYGD*, *CRYGB*, *CYP1B1*, *EFEMP1*, *FAM161A*, *IFT172*, *KCNJ13*, *MERTK*, *PIKFYVE*, *PRSS56*, *RAB3GAP1*, *SAG*, *SNRNP200*, *TTC21B*, *ZNF513*, *CHMP4B*, *IDH3B*, *PRPF6*, *SLC4A11*, *VSX1*, *CRYAA*, *LSS*, *CRYBA4*, *CRYBB1*, *CRYBB2*, *CRYBB3*, *TIMP3*, *ARL6*, *BFSP2*, *CLRN1*, *CRYGS*, *FYCO1*, *GNAT1*, *IMPG2*, *OPA1*, *RHO*, *SLC7A14*, *SOX2*, *CNGA1*, *CYP4V2*, *LRAT*, *LRIT3*, *PDE6B*, *PITX2*, *PROM1*, *RAB28*, *SLC25A4*, *TENM3*, *WFS1*, *GRM6*, *PDE6A*, *WDR36*, *COL11A2*, *ELOVL4*, *EYS*, *FOXC1*, *GCNT2*, *GUCA1A*, *IMPG1*, *LCA5*, *MAK*, *PRPH2*, *RIMS1*, *TULP1*, *AGK*, *IMPDH1*, *KLHL7*, *OPN1SW*, *RP9*, *SHH*, *TSPAN12*, *ADAM9*, *C8orf37*, *CNGB3*, *GDF6*, *HGSNAT*, *RP1*, *RP1L1*, *KCNV2*, *PRPF4*, *TDRD7*, *TOPORS*, *CACNA1F*, *CHM*, *CHRDL1*, *FRMD7*, *GPR143*, *NDP*, *NHS*, *OFD1*, *OPN1LW*, *OPN1MW*, *RP2*, *RPGR*, *RLBP1*, and *RS1*. The sequencing results were processed using the «NGS-Data» software v.1.1 [13].

Variant validation and biallelic status verification were carried out using direct automated Sanger sequencing.

The copies of *RPE65* exons 1–14 was analyzed with quantitative MLPA using an MRC-Holland SALSA MLPA probemix P221 LCA mix-1 kit. The results were evaluated in Coffalyser (MRC Holland, Netherlands) (https://www.mlpa.com).

The detected alterations in the *RPE65* gene were named according to the international HGVS nomenclature (http://varnomen.hgvs.org/recommendations/DNA, v.20.05) using the reference cDNA sequence presented on the NCBI portal (http://www.ncbi.nlm.nih.gov/nuccore): NM_000329.3.

The clinical significance of previously non-described nucleotide sequence variants was evaluated based on the Russian MPS data interpretation guidelines [14].

Written informed consent was obtained from the legal representatives of all patients under 18 years of age. Written informed consent from patients over 18 years of age was obtained. This study was conducted according to the guidelines of the Declaration of Helsinki and approved by the Ethics Committee of the Research Centre for Medical genetics.

## 3. Results

During this study, we established a molecular genetic diagnosis for 474 patients. The diagnosis was considered to be confirmed if (a) two pathogenic or likely pathogenic variants that have not been described previously in a cis state in genes with the autosomal recessive inheritance type were detected; a pathogenic or likely pathogenic variant was detected in a homozygous or hemizygous state; (b) a pathogenic or likely pathogenic variant was detected in a heterozygous state in a gene with the autosomal dominant inheritance type; (c) a pathogenic or likely pathogenic variant was detected in a hemizygous state in a gene with the X-linked inheritance type.

Among patients with a confirmed diagnosis, in 25 cases, the disorder was caused by biallelic mutations in the *RPE65* gene.

Four patients with RP caused by previously detected pathogenic variants in the *RPE65* gene were referred to our laboratory for segregation analysis. The trans state of the variants was confirmed for all four probands.

Thus, segregation analysis and confirmation of biallelic status were carried out for 29 patients with detected pathogenic or likely pathogenic variants in the *RPE65* gene (Figure 1). The patients’ age at the moment of their molecular genetic diagnosis of “RPE65-associated retinopathy” varied from 4 months to 50 years (Table 1).

We detected 26 different variants in the *RPE65* gene on 58 chromosomes; nine variants were novel—(c.1450+1G>A, c.1128+1G>A, c.897C>A/p.(Tyr299*), c.725G>T/p.(Ser242Ile) c.230dup/p.(Thr78Hisfs*10), c.1330C>T/p.(Pro444Ser), c.595_596delAAinsT p.(Asn199Phefs*9), c.1565T>A/p.(Ile522Asn), and c.503T>A/p.(Leu168His))—and two were previously described by us [15] (Figure 2, Table 2).

We detected 17 missense replacements in the *RPE65* gene: (c.272G>A/p.(Arg91Gln), c.65T>C/p.(Leu22Pro), c.1024T>C/p.(Tyr342His), c.1451G>A/p.(Gly484Asp), c.271C>T/p.(Arg91Trp), c.725G>T/p.(Ser242Ile), c.1307G>A/p.(Gly436Glu), c.746A>G/p.(Tyr249Cys), c.1330C>T/p.(Pro444Ser), c.1565T>A/p.(Ile522Asn), c.1451G>T/p.(Gly484Val), c.1340T>C/p.(Leu447Pro), c.982C>T/p.(Leu328Phe), c.617T>C/p.(Ile206Thr), c.118G>A/p.(Gly40Ser), c.503T>A/p.(Leu168His), c.1249G>C/p.(Glu417Gln)). We also detected four splice site mutations (c.11+5G>A, c.1450+1G>A, c.1128+1G>A, c.1451-G>A), three nonsense replacements (c.304G>T/p.(Glu102*), c.370C>T/p.(Arg124*), c.897C>A/p.(Tyr299*)), and two variants leading to a frameshift and the formation of a premature stop codon (c.230dup/p.(Thr78Hisfs*10) and c.595_596delAAinsT/p.(Asn199Phefs*9)) (Figure 3).

Two non-related patients (№8 and №19) had the c.304G>T/p.(Glu102*) variant in a homo-/hemizygous state, detected using MPS. When determining the zygosity, we discovered that one of each patient’s parents did not have the c.304G>T variant (in the first case, the variant was absent in the mother, and in the second case, in the father). The number of copies of *RPE65* exons 1 to 14 was analyzed using quantitative MLPA for both probands and their parents. Two copies of the *RPE65* gene were detected in the parents and probands themselves. In both cases, the relation was confirmed via an analysis of the microsatellite markers from 12 different chromosomes. To prove the hypothesis of uniparental isodisomy, we carried out an analysis using polymorphic markers from chromosome 1. As a result, in the first family, we established that at markers D1S450, D1S2663, and D1S189, the proband does not have maternal alleles, which signifies paternal uniparental disomy in the region containing the *RPE65* gene. In the second family, at markers D1S2663, D1S450, and D1S502, the proband did not have a paternal allele (Figure 4). Thus, in two families, the probands had a c.304G>T/p.(Glu102*) variant in a homozygous state because of uniparental isodisomy. The patients did not have any phenotypic abnormalities other than LCA, confirming previous works showing the absence of imprinted genes on chromosome 1 that have a significant effect on the phenotype [27,28]. A case of RPE65-associated retinopathy due to paternal uniparental isodisomy on chromosome 1 was first described by Thompson, D.A. et al. in 2002 [27]. Later, another case of RPE65-associated retinopathy caused by maternal uniparental isodisomy was described in the literature [28]. Moreover, multiple cases of various retinitis pigmentosa types were reported to be a result of alterations in other genes and uniparental isodisomy on chromosome 1: in the *ABCA4* gene [29], in the *USH2A* gene [30], and in the *CRB1* gene [31]. The established inheritance mechanism plays an essential role in medical genetic counseling for all family members.

## 4. Discussion

In the current study, we established a molecular genetic diagnosis of “RPE65-associated retinopathy” in 5.3% of patients with a confirmed IRD diagnosis. The obtained data mostly correspond well with the literature, according to which the prevalence of RPE65-associated retinopathy among patients with different retinitis pigmentosa types (confirmed via molecular genetic means) in Europe and North America varied from 3% in the USA and Spain to 9.98% in the Netherlands [32]. However, the prevalence of RPE65-associated retinopathy in Denmark and India differs from the worldwide average, reaching 16% and 16.6%, respectively [12,33].

Missense replacements were the most common nucleotide sequence variants in the current study, comprising 65% (17), followed by variants affecting the splice site—15% (4), then nonsense replacements—12% (3), then deletions and insertions leading to a frameshift and the formation of a premature stop codon—8% (2) (Figure 2). Thus, the obtained results confirm the data presented in previous studies: missense replacements in the *RPE65* gene are a common cause of RPE65-associated retinopathies [11,34].

According to the data presented in the literature, the most common pathogenic variants worldwide were c.271C>T/p.(Arg91Trp) (detected at 100 out of 864 pathogenic alleles (11.6%)), c.1102T>C/p.(Tyr368His) (69 out of 864 (8%)), and c.11+5G>A (62 out of 864 (7.2%)) [12]. Only two of these variants were detected in patients from the Russian Federation: c.271C>T and c.11+5G>A, being repeated, but not major. The prevalence of c.271C>T and c.11+5G>A in Russian patients with RPE65-associated retinopathy was 3.45% and 5.17%, respectively. The most common pathogenic variants encountered in the current study were c.304G>T/p.(Glu102*)—21.2%, c.370C>T/p.(Arg124*)—12%, and c.272G>A/p.(Arg91Gln)—8.6% of all affected chromosomes.

The most common variant in the examined cohort, c.304G>T/p.(Glu102*), was first described by Dharmaraj, S R et al. in 2000 in patients with LCA from the USA. In our cohort, this variant was detected in nine probands on thirteen chromosomes, with eleven of them being non-related (21.2%). It was initially described as major in the Netherlands [12]; however, it was never as common as in the current study. Patients carrying the c.304G>T variant were referred from various regions of the Russian Federation (Kursk, Irkutsk, Rostov, Yekaterinburg, Stavropol, Magadan); one of them was born in Turkmenistan and was ethnically Turkmen. All patients except for him were Russian.

A pathogenic variant, c.370C>T (p.Arg124*), was detected on seven chromosomes (12%). Two patients had the variant in a homozygous state, with three in a compound heterozygous state. This variant was initially described in 1998 by Morimura H et al. in patients from the USA and Canada [4]. It was present in various populations worldwide, never being major or common [35,36,37].

A nucleotide sequence variant, c.272G>A, leading to a p.(Arg91Gln) missense replacement and previously described as pathogenic, was detected on five chromosomes. It was initially described in 2000 by Thompson D.A. et al. in a compound heterozygous state [17]. A study by Philp AR et al. shows that the isomerase activity of RPE65 molecules with the p.(Arg91Gln) replacement comprised less than 6% of wild-type RPE65 activity [38]. The prevalence of this variant was 8.6%. It has been detected in Brazilian and Chinese patients [36,39]. This variant was also common in the study by Sallum JMF. et al.; however, its prevalence was significantly lower: 1.1%.

Variants c.11+5G>A and c.65T>C/p.(Leu22Pro) were detected on three chromosomes, each with the same prevalence of 5.17%.

The c.11+5G>A variant was detected in three patients in a compound heterozygous state. In the cohort of patients with RPE65-associated retinopathy from the USA and Europe, this variant was the most common, with a prevalence of 22.5%. The authors proved at least two independent origins of this variant [17]. The c.11+5G>A variant has been described multiple times as pathogenic in various worldwide populations; however, there were no more data on its prevalence among other alterations in the *RPE65* gene [22,37,40]. A study by Vázquez-Domínguez I. et al. shows that the c.11+5G>A variant does not affect splicing as expected according to prediction programs, but leads to a significant decrease in the expression of RPE65, specific for the pigment epithelium, with an undetermined mechanism [41].

The c.65T>C/p.(Leu22Pro) variant was detected in two patients in a homozygous and compound heterozygous state. It was initially described in 1998 in a compound heterozygous state in a patient with a “mild” disease course [19]. The p.(Leu22Pro) replacement was located in the inactive center of the enzyme. A study by Jin M et al. shows that the isomerase activity of RPE65 with the p.(Leu22Pro) mutation increases significantly at 30 °C. The researchers believe that a combination of a peroral intake of 4-phenylbutyrate (PBA) and a low-temperature eye mask provides “protein regeneration therapy”, which may increase the effectiveness of gene therapy [42].

It is worth noting that the five pathogenic variants listed above comprise more than half of all mutations in the *RPE65* gene in the examined cohort (31/58, 53.4%).

Six variants—c.1024T>C, c.1450+1G>A, c.1451G>A, c.271C>T, c.897C>A, and c.1128+1G>A—were detected on two chromosomes (3.45%) each. The c.1024T>C variant was detected in a compound heterozygous state, and others in a homozygous state.

The remaining 15 mutations, c.725G>T, c.230dup, c.1307G>A, c.746A>G, c.1330C>T, c.595_596delAAinsT, c.1565T>A, c.1451G>T, c.1451-G>A, c.1340T>C, c.982C>T, c.617T>C, c.118G>A, c.503T>A, and c.1249G>C, were detected once each.

The previously non-described variants—c.1128+1G>A and c.1450+1G>A—were located in the canonic splice site; c.230dup and c.595_596delAAinsT led to a frameshift with the formation of a premature stop codon, while the c.897C>A/p.(Tyr299*) variant led to the formation of a premature translation termination site in codon 299 (PVS1). These nucleotide sequence variants were not registered in gnomAD (The Genome Aggregation Database) control cohorts (PM2). The c.230dup variant was detected in a compound heterozygous state with a previously described pathogenic variant (PM3). Thus, according to the data interpretation guidelines [14], the detected nucleotide sequence variants should be regarded as likely pathogenic.

All previously non-described missense variants were also classified as likely pathogenic according to the data interpretation guidelines [14]: PM2—the variants were not registered in the gnomAD (The Genome Aggregation Database) control cohorts; PM3—the variants were detected in a compound heterozygous state with variants previously described as pathogenic; PP3—the pathogenicity prediction programs MutationTaster [43], FATHMM [44], LRT [45], and DEOGEN2 [46] evaluated these variants to be pathogenic; PP2—pathogenic missense variants in the *RPE65* gene are a common cause of RPE65-associated retinopathy [11,29].

We did not note any “hot” exons in the *RPE65* gene: pathogenic variants were detected in nine different exons—every exon except 1 and 11. Exons 4, 6, 12, and 14 had three variants each; exons 3 and 9 had two variants each; and the remaining exons had one pathogenic variant each (Figure 1). The location of the mutation in the gene has no influence on the effectiveness of target therapy.

In 2017, the Food and Drug Administration (FDA) approved a medication for gene therapy in patients with autosomal recessive retinitis pigmentosa caused by biallelic pathogenic variants in the *RPE65* gene. The therapy is based on a recombinant adeno-associated viral vector (AAV2) carrying a functioning copy of normal human *RPE65*. Gene therapy using AAV2 injects a normal copy of the gene into the cell, not repairing or deleting the defective gene [47]. When placed in sustainable cells of pigment epithelium via a singular subretinal injection, the medication provides the potential for vision cycle reparation [48]. Thus, an accurate molecular diagnosis is essential for the approval of target therapy.

All patients with confirmed biallelic variants in the *RPE65* gene, as well as the patient with uniparental isodisomy, are suitable for this target therapy.

The current study shows that pathogenic variants in the *RPE65* gene contribute significantly to the pathogenesis of IRDs and comprise 5.3% of all patients with a confirmed molecular genetic diagnosis. The distinctive specificity of the spectrum was the high prevalence of a single variant—c.304G>T/p.(Glu102*)—of 21.2% among all probands. The most common pathogenic variants in patients with RPE65-associated retinopathy from the Russian Federation were c.304G>T/p.(Glu102*), c.370C>T/p.(Arg124*), and c.272G>A/p.(Arg91Gln), comprising 41.8% of all affected chromosomes. Two out of the three most common pathogenic variants worldwide—c.271C>T and c.11+5G>A—were detected in the Russian cohort on two and three non-related chromosomes, respectively, but were not the most common.

Seeing as the sooner the patient undergoes treatment, the more cells react to therapy, and that the presence of biallelic variants in the *RPE65* gene is a necessary condition for gene therapy, molecular genetic diagnostics need to be carried out as soon as possible in cases of suspected RP. The current study allowed us to form a cohort for target therapy for the disorder; such therapy has already been carried out for some patients.

## Figures and Tables

**Figure 1 genes-14-02056-f001:**
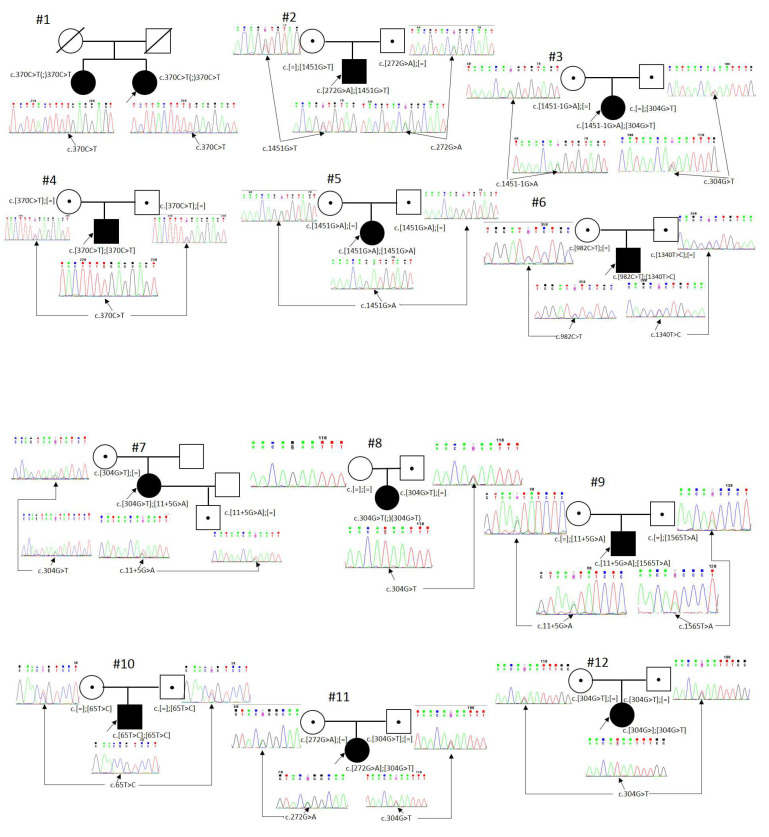
Segregation analysis. Black—affected, white—unaffected, dot—carrier, square—male, circle—female, arrow—proband, slash—deceased. #1, #2, …, #29—Patient ID.

**Figure 2 genes-14-02056-f002:**
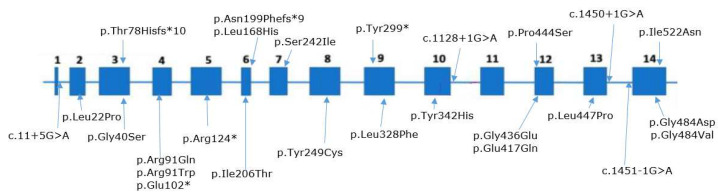
The structure of the *RPE65* gene with the detected nucleotide sequence variants. Previously non-described variants are placed above the gene structure.

**Figure 3 genes-14-02056-f003:**
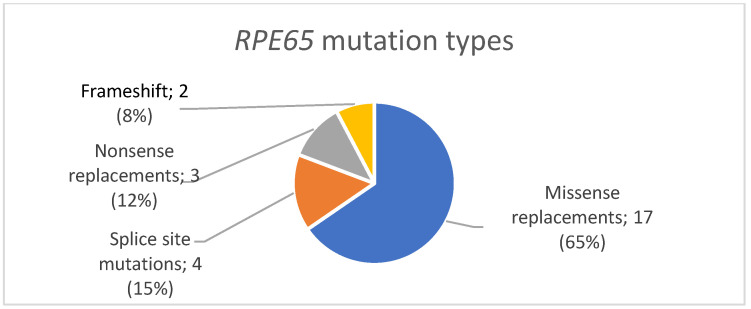
The distribution of *RPE65* mutation types.

**Figure 4 genes-14-02056-f004:**
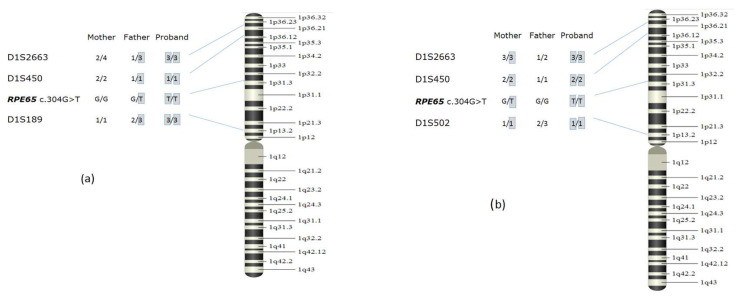
Microsatellite marker analysis on chromosome 1 in families with uniparental isodisomy. (**a**) Paternal uniparental isodisomy, patient #8; (**b**) maternal uniparental isodisomy, patient #19.

**Table 1 genes-14-02056-t001:** Patients and genotypes.

Patient ID	Age at the Moment of Molecular Genetic Diagnosis	Sex	Place of Residence	Ethnic Group	Genotype	Nystagmus	Nyctalopia	Snellen BCVA		Central Foveal Retinal Thickness, µm		Color Vision Anomaly	ffERG
								OD	OS	OD	OS	OU	OU
1	50 years	f	Chita region	Buryat	c.[370C>T]; [370C>T]	Yes	Full	0.001	0.001	80	90	MCh	ND
2	5 years	m	Moscow region	Belarusian	c.[1451G>T]; [272G>A]	Yes	Full	0.16	0.2	202	185	Deu	Ext
3	5 years	f	Kursk region	Russian	c.[1451-1G>A]; [304G>T]	No	inc	0.2	0.2	134	129	No	ND
4	26 years	m	Buryatia	Buryat	c.[370C>T]; [370C>T]	Yes	Full	0.01	0.01	90	102	MCh	ND
5	13 years	f	Dagestan	Lezgin	c.[1451G>A]; [1451G>A]	Yes	Full	0.01	0.01	137	124	Deu	Ext
6	12 years		Yamalo-Nenets Autonomous Okrug	Russian	c.[982C>T]; [1340T>C]	No	inc	0.3	0.16	230	247	No	SubN
7	31 year	f	Irkutsk region	Russian	c.[304G>T]; [11+5G>A]	Yes	Full	0.05	0.05	134	133	MCh	ND
8	6 years	m	Rostov region	Russian	c.304G>T(;) (304G>T)	No	inc	0.4	0.4	153	158	Tri	Ext
9	3 years	m	Leningrad region	Russian	c.[11+5G>A]; [1565T>A]	Yes	Full	0.2	0.2	176	176	Tri	SubN
10	9 years	m	Dagestan	Kumyk	c.[65T>C]; [65T>C]	Yes	inc	0.6	0.7	170	185	AnT	SubN
11	6 years	f	Sverdlovsk region	Russian	c.[272G>A]; [304G>T]	Yes	inc	0.05	0.05	182	221	Dich	SubN
12	36 years	f	Moscow region	Turkmen	c.[304G>]; [304G>T]	Yes	Full	0.001	0.001	100	90	MCh	Ext
13	28 years	m	Moscow region	Russian	c.[1330C>T]; [595_596delAAinsT]	Yes	Full	0.15	0.05	204	195	MCh	ND
14	24 years	m	Stavropol region	Russian	c.[304G>]; [304G>T]	Yes	Full	0.01	0.01	199	216	Dich	ND
15	18 years	m	Bashkortostan	Bashkir	c.[271C>T]; [271C>T]	Yes	Full	0.3	0.2	165	158	AnT	ND
16	12 years	m	Magadan Region	Russian	c.[304G>T]; [370C>T]	Yes	inc	0.2	0.2	157	152	AnT	SubN
17	12 years	m	Sverdlovsk region	Tajik	c.[1450+1G>A]; [1450+1G>A]	Yes	inc	0.1	0.05	213	271	MCh	SubN
18	11 years	m	Stavropol region	Dargin	c.[897C>A]; [897C>A]	Yes	Full	0.05	0.2	234	215	Tri	SubN
19	4 years	m	Bryansk region	Russian	c.304G>T(;) (304G>T)	Yes	Full	0.01	0.01	144	132	AnT	Ext
20	4 months	f	Smolensk region	Russian	c.[503T>A]; [304G>T]	No	inc	sv	sv	-	-	-	-
21	1 year	m	Vologda region	Interethnic—Uzbek-Russian	c.[272G>A]; [725G>T]	Yes	inc	sv	sv	-	-	-	-
22	5 years	f	Tuva region	Tuvan	c.[370C>T]; [1024T>C]	Yes	inc	0.35	0.4	203	200	Deu	Ext
23	18 years	f	Moscow region	Russian	c.[230dup]; [272G>A]	Yes	Full	0.1	0.16	271	245	AnT	ND
24	11 years	f	Leningrad region	Tajik	c.[1128G>A]; [1128G>A]	Yes	Full	0.2	0.1	213	210	Deu	Ext
25	28 years	m	Omsk region	Russian	c.[617C>T]; [118G>A]	Yes	Full	0.1	0.1	140	120	MCh	ND
26	4 years	f	Moscow region	Russian	c.[1307G>A]; [746G>A]	Yes	Full	0.2	0.2	170	168	AnT	Ext
27	13 years	f	Tuva region	Tuvan	c.[370C>T]; [1024T>C]	No	Full	0.3	0.3	172	180	MCh	SubN
28	43 years	m	Ivanovo region	Russian	c.[272G>A]; [11+5G>A]	Yes	Full	0.001	0.001	186	134	MCh	ND
29	7 years	f	Buryatia	Kumyk	c.[1249G>C]; [65T>C]	Yes	Full	0.4	0.4	189	179	AnT	ND

Abbreviations: f—female, m—male, inc—incomplete, sv—Subject vision, MCh—monochromasia, Deu—deuteranopia, Tri—tritanopia, AnT—anomalous trichromat, Dich—dichromat, ND—non-detectable, Ext—Extinguished, SubN—subnormal.

**Table 2 genes-14-02056-t002:** Detected variants in the *RPE65* gene.

№	Variant	Effect	Exon/Intron	№ of chr.	Prevalence, %	Allele Frequency in gnomAD	References/Pathogenicity Criteria
1	c.304G>T	p.(Glu102*)	ex 4	13 (11)	21.2	0.00003580	[16]
2	c.370C>T	p.(Arg124*)	ex 5	7	12.0	0.00005674	[4]
3	c.272G>A	p.(Arg91Gln)	ex 4	5	8.6	0.00004600	[17]
4	c.11+5G>A	splicing	in 1	3	5.17	0.000078	[18]
5	c.65T>C	p.(Leu22Pro)	ex 2	3	5.17	0.000028	[19]
6	c.1024T>C	p.(Tyr342His)	ex 10	2	3.45	n/a	[15]
7	c.1450+1G>A	splicing	in 13	2	3.45	n/a	PVS1, PM2
8	c.1128+1G>A	splicing	in 10	2	3.45	n/a	PVS1, PM2
9	c.1451G>A	p.(Gly484Asp)	ex 14	2	3.45	0.000008047	[20]
10	c.271C>T	p.(Arg91Trp)	ex 4	2	3.45	0.000053	[4]
11	c.897C>A	p.(Tyr299*)	ex 9	2	3.45	n/d	PVS1, PM2
12	c.725G>T	p.(Ser242Ile)	ex 7	1	1.72	n/d	PM2, PP3, PP2, PM3
13	c.230dup	p.(Thr78Hisfs*10)	ex 3	1	1.72	n/d	PVS1, PM2,PM3
14	c.1307G>A	p.(Gly436Glu)	ex 12	1	1.72	n/d	[17]
15	c.746A>G	p.(Tyr249Cys)	ex 8	1	1.72	0.00001773	[21]
16	c.1330C>T	p.(Pro444Ser)	ex 12	1	1.72	n/d	PM2, PP3, PP2, PM3
17	c.595_596delAAinsT	p.(Asn199Phefs*9)	ex 6	1	1.72	n/d	PVS1, PM2
18	c.1565T>A	p.(Ile522Asn)	ex 14	1	1.72	n/d	PM2, PP3, PP2, PM3
19	c.1451G>T	p.(Gly484Val)	ex 14	1	1.72	0.00001207	[22]
20	c.1451-G>A	splicing	in 14	1	1.72	0.000004024	[23]
21	c.1340T>C	p.(Leu447Pro)	ex 13	1	1.72	n/d	[15]
22	c.982C>T	p.(Leu328Phe)	ex 9	1	1.72	0.039	[24]
23	c.617T>C	p.(Ile206Thr)	ex 6	1	1.72	0.000012	[25]
24	c.118G>A	p.(Gly40Ser)	ex 3	1	1.72	0.000028	[4]
25	c.503T>A	p.(Leu168His)	ex 6	1	1.72	n/d	PM2, PP3, PP2, PM3
26	c.1249G>C	p.(Glu417Gln)	ex 12	1	1.72	0.000004	[26]

## Data Availability

Not applicable.

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
