# Peer review of "A Molecular Genetic Analysis of RPE65-Associated Forms of Inherited Retinal Degenerations in the Russian Federation"

_genes, 2023, doi:10.3390/genes14112056_

Round 1

Reviewer 1 Report

Comments and Suggestions for Authors

The study by Stepanova and co-workers, investigates genetic alterations underlying RPE65-related retinal degenerations in 1053 patients from the Russian Federation. The authors analysed a panel of 211 genes by next generation sequencing and were able to establish a genetic diagnosis in 474 cases, of which 25 carried RPE65 variants. The analysis identified 26 different RPE65 variants, including nine mutations claimed by the authors to be previously unidentified.

General comments.

This manuscript is interesting and contributes to characterizing RPE-related retinal degenerations in the Russian population. In addition, it expands the spectrum of pathogenic RPE65 variants. However, there are important concerns that must be addressed.

Main comments

1.- The authors should explain how the diagnosis of inherited retinal degeneration was established.

2.- Inclusion and exclusion criteria should also be described.

3.- Provide demographic and clinical information about the cohort.

4.-To evaluate the genotype-phenotype relationship, additional demographic and clinical data, including the clinical diagnosis, of the 26 RPE65 patients must be provided.

5.- To support the pathogenicity of the identified variants, it is important to show the segregation analysis and the electropherograms confirming the variants by Sanger sequencing.

6.- The identification of two cases of uniparental isodisomy of chromosome 1 must be described in the results section, not in the discussion. Please describe and discuss whether these two patients present imprinting-related conditions.

7- Indicate the position of the RPE65 gene relative to the four markers used to analyse isodisomy.

Minor comments.

8.- To facilitate reading, the list of variants described in lines 140-150 and 249-254, should be simplified.

9.- Tables. Change “cDNA localization” to “cDNA variant” and “effect” to “protein variant”.

10. Table 10. Provide the pathogenicity criteria for the described variants in addition to the reference.

11.- Figure 3. Translate text in Russian to English.

12.- Check the format of some citations, see for instance lines 180 and 269.

13.- The first sentence in line 123 seems to be incomplete, please revise it for clarity.

14.- Provide additional information on the “Ophthalmo gene panel”. Is it commercial? What type of Sequencer was used?

Author Response

Response to Reviewer Comments.

Thank you very much for taking the time to review this manuscript.

According to your comments, the manuscript has been corrected and supplemented. All changes in the manuscript are highlighted in blue font.

Response 1, 2: [The authors should explain how the diagnosis of inherited retinal degeneration was established.  Inclusion and exclusion criteria should also be described.]

Thank you for pointing this out. Information on ophthalmological examination methods and criteria for selecting patients for the study has been added to the materials.

Response 3, 4: [Provide demographic and clinical information about the cohort. To evaluate the genotype-phenotype relationship, additional demographic and clinical data, including the clinical diagnosis, of the 26 RPE65 patients must be provided.]

The information on the demographic and clinical data of 29 patients with RPE65-dependent retinopathy was Table 1 adds.

Response 5: [To support the pathogenicity of the identified variants, it is important to show the segregation analysis and the electropherograms confirming the variants by Sanger sequencing]

Figure 1 contains segregation analysis data and electropherograms.

Response 6, 7: [The identification of two cases of uniparental isodisomy of chromosome 1 must be described in the results section, not in the discussion. Please describe and discuss whether these two patients present imprinting-related conditions.  Indicate the position of the RPE65 gene relative to the four markers used to analyse isodisomy.] We agree with this comment.

Cases of uniparental isodisomy were transferred to the results. Added information about the absence of conditions associated with imprinting. Figure 4  shows the location of the RPE65 gene (#8 between markers D1S450 and D1S189, #19 between markers D1S450 and D1S502).

In the methods I added information about the Ophthalmo panel and the sequencer.

I also tried to correct the remaining comments.

Additional clarifications: A big table (Table 1) and a big figure (Figure 1) have been added to the text of the manuscript. It would be better to add the table and figure as supplementary materials. If at this stage it is possible.

Reviewer 2 Report

Comments and Suggestions for Authors

In their manuscript, Stepanova and colleagues review the genetic causes of IRDs within the general population of the Russian federation, focusing on the RPE65 genotype, With gene therapy for RPE-65-related retinal dystrophies already approved and many more "Single gene -single protein" disorders being investigated with gene therapy being currently developed, this manuscript is both of interest and timely. The enthusiasm for this manuscript is lessened, however, by several recognizable weaknesses. 

1) The manuscript is essentially devoid of any clinical data. As known, many clinical considerations are relevant when a decision regarding gene therapy is made. For instance, what were the visual acuities of the patients? Level of their disability?

2) What were the results of the patients OCT imaging? As we know, gene therapy requires at least some intact photoreceptors in order to exert its effect. How many subjects had measurable ONL? Correlation with age and other factors?

3) Demographic data: there's no real demographic data regarding the patients other than their age. Were all participants of the same demographic background?

4) The manuscript is in need of professional editing services to improve the quality of its English

I thank the authors for the opportunity to review their work and wish them luck in their future endeavors. 

Comments on the Quality of English Language

 The manuscript is in need of professional editing services to improve the quality of its English

Author Response

Thank you very much for taking the time to review this manuscript.

According to your comments, the manuscript has been corrected and supplemented. All changes in the manuscript are highlighted in blue font.

Response 1) The manuscript is essentially devoid of any clinical data. As known, many clinical considerations are relevant when a decision regarding gene therapy is made. For instance, what were the visual acuities of the patients? Level of their disability?

2) What were the results of the patients OCT imaging? As we know, gene therapy requires at least some intact photoreceptors in order to exert its effect. How many subjects had measurable ONL? Correlation with age and other factors?

3) Demographic data: there's no real demographic data regarding the patients other than their age. Were all participants of the same demographic background?

4) The manuscript is in need of professional editing services to improve the quality of its English

Thank you for pointing this out. Information on ophthalmological examination methods and criteria for selecting patients for the study has been added to the materials.

The information on the demographic and clinical data of 29 patients with RPE65- associated retinopathy was Table 1 adds.

In the methods I added information about the Ophthalmo panel and the sequencer.

I also tried to correct the remaining comments.

Additional clarifications: A big table (Table 1) and a big figure (Figure 1) have been added to the text of the manuscript. It would be better to add the table and figure as supplementary materials. If at this stage it is possible.

Round 2

Reviewer 1 Report

Comments and Suggestions for Authors

The authors have responded to the questions raised in the previous review, and the quality of the paper has been improved. However, Figure 1 is not clear, and the format requires further improvement. The meaning of all symbols and colors in symbols must be explained. Please use conventional nomenclature to indicate the genotypes under the symbols. I also recommend separating pedigrees and electropherograms into two different figures

Author Response

Thank you for your careful work with our manuscript, your comments helped make our work much better. The drawings have been corrected in accordance with the nomenclature, the meaning of the symbols and colors in the symbols have been added.

Reviewer 2 Report

Comments and Suggestions for Authors

The authors have adequately addressed this reviewer's scientific comments. 

Comments on the Quality of English Language

I believe the article can benefit from professional scientific editing services 

Author Response

Thank you for your careful work with our manuscript, your comments helped make our work much better.